# Addressing obstetricians' awareness of compassion, communication, and self-care when caring for families experiencing stillbirth: Evaluation of a novel educational workshop using applied drama techniques

Karen McNamara[1], Aisling Smith[2], Brid Shine[3], Mairie Cregan[1,4], Lucia Prihodova[2], Ann O'Shaughnessy[2], Aisling Martin[3,5], Jenny Macdonald[6], Phil Kingston[6], Chris Fitzpatrick[3,5], Marita Hennessy[1,7]*, Keelin O'Donoghue[1,7]

1 Pregnancy Loss Research Group, Department of Obstetrics and Gynaecology, University College Cork, Cork, Ireland, 2 Royal College of Physicians of Ireland, Dublin, Ireland, 3 Coombe Women & Infants University Hospital, Dublin, Ireland, 4 Féileacáin, Stillbirth and Neonatal Death Association of Ireland, Cork, Ireland, 5 UCD Centre for Human Reproduction, Dublin, Ireland, 6 Abbey Theatre, Dublin, Ireland, 7 INFANT Research Centre, University College Cork, Cork, Ireland

* maritahennessy@ucc.ie

**Data Availability Statement:** Data cannot be shared publicly because it contains sensitive

## Abstract

### Introduction

Obstetricians describe feeling shocked and isolated following stillbirth. Few receive adequate training in how to care for bereaved parents or themselves. We developed a novel workshop for trainee obstetricians using applied drama techniques–in collaboration with the National Theatre of Ireland, the national training body for obstetricians and gynaecologists, and patient support groups–to teach obstetricians skills in communication and self-care around the time of stillbirth.

### Materials and methods

Five workshops, delivered January-May 2018, are the focus of this evaluation. Senior trainees in Obstetrics attended and completed a post-workshop evaluation questionnaire. Five-point Likert scales were used to assess participants' communication and support skills pre- and post- the workshop, and their views on pre-specified attributes needed when caring for families experiencing stillbirth and aspects of the workshop. Quantitative and qualitative data were analysed using descriptive statistics and content analysis, respectively.

### Results

39/59 (66%) workshop participants completed the questionnaires. Most had received no prior training in caring for families experiencing antenatal (31/39, 80%) or intrapartum (34/39, 87%) stillbirth. Following the workshop there was a significant improvement in trainee's level of confidence in breaking bad news, communicating clearly with the family when

information about the participants and they did not provide consent for public data sharing. The current approval by the Research Ethics Committee at the Royal College of Physicians of Ireland (RCPI RECSAF 71) does not include data sharing. A minimal data set could be shared by request for the sole purpose of replicating the present study. Data requests may be sent to the senior author (KOD; k.odonoghue@ucc.ie) or the Research Ethics Committee at the Royal College of Physicians of Ireland (research@rcpi.ie).

**Funding:** This study was completed in part in the Irish Centre for Fetal and Neonatal Translational Research (now known as The Irish Centre for Maternal and Child Health) and as such was in part funded by Science Foundation Ireland (12/RC/2722).

**Competing interests:** The authors have declared that no competing interests exist.

breaking bad news, recognising the emotional needs of the family, recognising their own emotional responses, and supporting their colleagues. Trainees were positive about the workshop content and delivery; 90% stated they would recommend it to a colleague.

## Discussion

Adequate, appropriate, and stimulating education and training in stillbirth care and self-care is clearly needed to improve patient care. Our findings demonstrate that this novel educational workshop using applied drama techniques–developed in collaboration with diverse stakeholders and underpinned by the views of parents and obstetricians who had experience of stillbirth–is an acceptable and appropriate way of training obstetricians in how to care for bereaved parents and/or to engage in self-care.

## Introduction

The death of a baby during pregnancy, labour or the neonatal period is profoundly upsetting for parents and families and the healthcare professionals involved [1–7]. Obstetricians describe feeling shocked, sad, isolated, afraid, angry and traumatised in the aftermath of a stillbirth [2–4, 8], which, for some, is considered the worst outcome of pregnancy for parents [2]. They can vividly recall in detail the minutiae of some of the stillbirths they have been involved with, bringing up particularly painful memories for some [2, 4]. Some also dwell on the specifics of the stillbirth after they have left the workplace [2]. Others can find it difficult to speak about these events with colleagues or family members [2].

Despite the considerable impact that involvement in a stillbirth has on obstetricians, few—if any—receive adequate training in how to care for parents in the aftermath of a stillbirth, or in self-care skills [2, 3, 6, 9–12]. There is some evidence to suggest that obstetricians who have received adequate education in stillbirth care are less likely to report feeling guilty and afraid of litigation in the aftermath of their involvement with a stillbirth [12]. Lack of training in stillbirth management can impact negatively on the care that obstetricians give to bereaved parents [12].

In Ireland, the importance of empathy and compassion in bereavement care has been highlighted in the National Maternity Strategy, National Standards for Safer Better Maternity Services and National Standards for Bereavement Care in the maternity services [13–15]. There has also been increasing concern expressed by national professional training bodies about the effects of traumatic work-related experiences on clinicians [16].

Directly inspired by personal advice given to one author (CF) by Oscar Laureate, the late Sir Alec Guinness (Box 1), and the outreach work of the National Theatre of Ireland, the Abbey Theatre (designed by PK), the Royal College of Physicians of Ireland (RCPI) were approached (by CF and PK) to develop an innovative educational workshop for trainees in Obstetrics and Gynaecology, in collaboration with the Institute of Obstetricians and Gynaecologists (the national training body for all obstetricians and gynaecologists in Ireland) and patient support groups *Féileacáin* and *Patient Focus*. The workshop was entitled 'Bereavement in the Maternity Services: An Approach to Caring and Coping for Clinicians using Applied Drama Techniques'. Techniques and principles from theatre have been used effectively in medical education regarding difficult-to-teach communication-related concepts and skills across a variety of areas [17–21].

A Steering Group, comprising representatives from each of the partner organisations, was assembled to provide guidance and support in the development and implementation of the

## Box 1. Quotation.

"*Doctors need to be like actors. They need to see the world through the eyes of their patients–like actors see the world through the eyes of their characters. They need to say their lines with great sincerity. They also need to be able to switch off afterwards*"

Sir Alec Guinness

project. *Féileacáin* is a not-for-profit organisation in Ireland, that provides support to parents who are affected by the death of a baby either in pregnancy or shortly after. Until July 2018, Patient Focus was one of the country's leading patient advocacy services—before its merging with another national advocacy service, Sage Advocacy. There has been increased emphasis on patient involvement in medical education in recent years; however there is a dearth of studies focusing on patient involvement in postgraduate education [22].

The aim of this study was to evaluate this novel workshop, focusing in particular on the impact on: communication skills, recognition of emotional needs of the bereaved family, recognition of participant's own needs when caring for bereaved families, self-efficacy in breaking bad news, and whether or not participants developed any new skills that could be translated to the clinical environment. It should be noted at this point that the workshop was not a training intervention focused on increasing participants' knowledge and skills regarding breaking bad news; however, insights and skills acquired during the workshop could potentially enhance participants' self-efficacy in breaking bad news.

## Materials and methods

### Workshop development and delivery

The incorporation of the views of parents and obstetricians who had experience of stillbirth were central to the development of this educational intervention. Invitations were issued to parents who had experienced a late pregnancy stillbirth in the preceding five years to be interviewed and narrate their experience of loss. Parents were identified and contacted by the two partner support organisations. Obstetricians who had been involved in caring for families who experienced a stillbirth over the same time period were also invited for interview to share their experiences; facilitated through the Institute of Obstetricians and Gynaecologists. Interviews were conducted by an experienced bereavement and loss midwife (BS) and were audio-recorded and transcribed verbatim.

Using the blended, anonymised narratives from interviews with ten parents (two fathers and eight mothers, including two interviews with couples) and six obstetricians, performance specialists from The Abbey Theatre (JM and PK), in association with educational specialists (AS and AOS) and consultant trainers in the RCPI (CF, AM and KOD) and *Féileacáin* (MC), developed an applied drama workshop for trainees in Obstetrics and Gynaecology. The performance specialists–JM (writer, performer, director and facilitator) and PK (Community and Education Manager, Abbey Theatre; actor, writer, director and facilitator)—approached the narratives of parents and obstetricians, as both artists and applied theatre practitioners, with the understanding that applied theatre facilitation is an art form. In the rehearsal rooms of the Abbey Theatre, they both worked with the narratives by 'casting the net, drawing it in, taking a breath' and writing it down [23], acting their responses out, rather than adopting a conventional qualitative analytic approach. The narratives allowed them to immerse themselves in,

and embody, the worlds of obstetricians and bereaved parents, with a view to harnessing the energy created in this rehearsal space to emanate during workshops. As artists, JM and PK identified resonant images and tropes across the stories told in the interviews. They found patterns in how bereaved parents articulated their trauma and how the health professionals reacted personally to incidents of stillbirth. This distinctive imagery helped in designing the workshop, which at one point had the working title of *Three Rooms* to reflect the three rooms mothers often ended up being asked to occupy. The imagery also helped them deepen their engagement with the trainees in the workshops. Key themes integrated from the narratives into the workshops were: clarity of language, equal stance, and space to react.

The specific learning aims and objectives of the workshops were developed and agreed by the Steering Group, and are presented in Table 1. Key objectives included helping trainees to maintain a sense of authenticity whilst doing the same thing (i.e. breaking bad news) over and over again, develop their coping skills, and manage the balance of professionalism and developing a positive patient relationship. While the aims and objectives were the same for each workshop, the exact content of each workshop differed somewhat each time, as trainees differed in terms of their experiences and inputs/level of interaction, and facilitators also evolved after engaging in and reflecting on each workshop.

Through the use of applied drama techniques, the emotional experiences of stillbirth were explored with trainees through a number of writing, acting, movement and reflective activities (see Box 2). Trainees engaged in kinaesthetic learning with the facilitators, reconnecting them with their bodies and emotional memories. Silence, presence and authenticity were emphasised throughout; trainees were also encouraged to use other ways to communicate than the spoken word as 'words can go hollow'. As such, the workshops did not 'teach acting', rather the focus was on building and fostering empathy, amongst other skills.

The morning sessions focused on the lived experience of bereaved parents, while the afternoon sessions focused on the clinician, investigating who they are and what their role is. Trainees were briefed at the outset of each workshop about the nature of the workshops and the potential for distress. They were reassured that it was fine to leave the space for a time if a particular activity was upsetting for them, and to seek any necessary supports. On a few occasions, trainees approached facilitators on breaks to say they might find the material upsetting. In

**Table 1. Workshop learning aims and objectives.**

| During the workshop trainees will: | Following this workshop, the trainee will be able to: |
|---|---|
| • Learn physical and vocal skills to increase self-awareness and support a sense of authenticity and presence<br>• Gain understanding of the importance of self-awareness in communication<br>• Increase their understanding the connection between self-awareness, presence, authenticity and compassion<br>• Apply a practical understanding of self-awareness to achieve authenticity in communication<br>• Explore the role of language in empathy<br>• Use language precisely<br>• Learn to balance compassion and resilience<br>• Reconcile any tension between the clinical and the humane | • Demonstrate a sense of presence when needed to break tragic news<br>• Connect to parents who have experienced stillbirth<br>• Explore their own emotional responses to better respond to grieving parents<br>• Choose language, stance and gesture appropriate to the situation<br>• Demonstrate an increased awareness of how you say what you say to parents who have experienced stillbirth<br>• Actively listen and appropriately respond to communicate compassionately with parents<br>• Build the rigour and resilience to keep caring with each new encounter<br>• Reconcile the philosophical and practical paradox of being authentic and caring repeatedly |

---

### Box 2. Workshop activities.

**Movement work**

With added mindfulness to tease out the layers of communication in the doctor/patient relationship

**Perspective-taking**

Contemplating quotations from actual patients to explore empathy, the same way an actor would approach a text

**Tableaux creation**

Thought shadowing where you make a freeze frame of a moment in a therapeutic encounter and then have the trainees voice the thoughts and feelings going through their heads

**Scene creations**

Creating short scenes whose theatricality is emphasised using various prompts which add a layer of artificiality so that the form enhances the emotional content

**Rigour and repetition**

Recreating the rehearsal room to create an objectivity about difficult content while building the resilience to keep 'meeting' that content emotionally

---

these cases, the facilitators reiterated that it was fine to take breaks and/or leave the room to allow space for their responses. See Table 2 for a full workshop programme.

Each workshop had a maximum of fourteen places, with a minimum of six trainees to facilitate group activities, and was six hours in duration. Five workshops were held between January and May 2018, one each month. Workshops were facilitated by JM and PK, their combined experience bringing trainees, initially uncomfortable with the idea of drama and performance, to the creation of full scenes by the end of a seven-hour day in an artistic atmosphere that was at once productive, creative, and supportive [23].

### Sample

All those on training schemes in Obstetrics and Gynaecology within the RCPI between January and May 2018 (n = 74) were invited, by direct email, to attend a workshop and participate in the evaluation. Participation was also encouraged by the National Specialty Director. It should be noted that although these were pilot workshops, they were included as a mandatory component of the RCPI's core Higher Specialist Trainee Obstetrics and Gynaecology curriculum [24].

### Data collection

A specific anonymised post-workshop evaluation questionnaire was designed by KMN, KOD, LP and AS as there was no pre-existing tool relevant to our specific research questions (see copy of questionnaire in S1 Appendix). Five-point Likert scales were used to assess participants' communication and support skills pre- and post- the workshop, as well as participants' views on attributes needed when caring for families experiencing stillbirth (self-awareness,

**Table 2. Workshop programme.**

| Time | Topic |
|---|---|
| 09:30–10.00 | Introduction<br>• Introduction to methodology<br>• Contract of confidentiality<br>• Physical warmup: 8 shake; Colombian hypnosis; group introductions |
| 10:00–11:30 | The physical basis<br>• Person-to-person<br>• Comfort/discomfort<br>• Reflective writing<br>• Reflective discussion with whole group<br>• Mirroring<br>• Statues<br>• Breathing/vocal exercises: Deep-breathing; singing<br>• Story swap<br>• Reflective discussion |
| 11:30–11:45 | Break |
| 11:45–12:50 | Staging one of three quotations (from bereaved parents) in groups of 3<br>• Groups present the small scenes to one another and give feedback. Scenes are repeated several times as a practice in returning to a repeated moment with authenticity<br>• Reflective discussion |
| 12:50–13:00 | Period of timed writing to reflect on morning |
| 13:00–14:00 | Lunch |
| 14:00–14:30 | Warm-Up<br>• Clown beach ball<br>• 'Sorry, I'm late.'–status: A table and chair are set up and one person has to enter the room, sit in the chair and apologise for being late |
| 14:30–14:45 | Interview<br>• In pairs, participants interview each other as if they were an actor going to play the role of a doctor |
| 14:45–15:45 | Creating Short Dramas<br>• In groups of three, participants are asked to reflect on a difficult situation where they didn't know what to do, and to then to make a short play about it but with specific parameters (one line of a song, a profound and unexpected silence, repetitive movement without sound, stillness, use a maximum of five words of text)<br>• Staging the work: Each group acts out their piece, facilitators—in collaboration with participants—then interrogate how effectively it communicates its message; plays are acted out at least three more times |
| 15:45–15:55 | Period of timed writing |
| 15:55–16:15 | Reflective discussion–on the day as a whole |
| 16:15–16:30 | Thanks and closing-Taking a bow |

Note: Applied drama projects are bespoke and not the creation of an off-the-peg methodology which can then be reproduced by anyone following the instructions. Any education organisation wanting to roll out a similar project would need to engage their own applied drama practitioners and reproduce the months long research period, articulate specific learning outcomes and then engage in a collaborative design process before engaging in similar activities. The workshop outline above was created and delivered by PK and JM using this process.

**Table 3. Internal consistency for each of the Likert scales.**

| Scale | Cronbach's alpha |
|---|---|
| Trainees' prior confidence in breaking bad news, communicating with bereaved families, recognising their own and the families' emotional needs, as well as their perceived abilities to support colleagues | 0.83 |
| Trainees perceived confidence in in breaking bad news, communicating with bereaved families, recognising their own and the families' emotional needs, as well as their perceived abilities to support colleagues after attending the workshop | 0.903 |
| Whether the content of the workshop met their needs for caring for bereaved parents | 0.82 |
| Overall assessment rating of the workshop practicalities | 0.938 |

authenticity, their own emotional responses, the role of verbal communication, the role of non-verbal communication, active listening, building of rapport, building resilience) and aspects of the workshop itself. Basic demographics were also obtained, along with questions regarding any training received in, or involvement in perinatal deaths. Free text questions were also included to enable participants to provide additional feedback on the workshops. Participants were asked to provide at least one positive comment and one point for improvement on the workshop. The questionnaire was administered, in-person and in hardcopy, by a representative of the RCPI (AS) at the end of each workshop.

## Data analysis

Descriptive statistics were used to examine the self-reported change in knowledge and attitudes that occurred as a result of attending the workshop. A paired-samples t-test with a significance level set at 0.05 was used to test for self-reported changes in the skills and attributes of the trainees following the workshop. Internal Consistency for each of the Likert scales was checked using Cronbach's alpha; all demonstrated good internal consistency, with Cronbach alpha scores greater than 0.7 (see Table 3). Open-ended responses were analysed using conventional content analysis [25].

## Ethical approval

Ethical approval was obtained from the Research Ethics Committee at the Royal College of Physicians of Ireland (RCPI RECSAF 71). Participants provided informed written consent prior to completing the evaluation questionnaire.

## Results

In total, over the course of the five workshops, 59 trainees participated (59/74; 80%). Of those who attended, 39/59 (66%) completed the workshop evaluation questionnaires. The main reason for non-participation in the evaluation was trainees leaving immediately after the workshop, or slightly earlier, due to other commitments. Most participants (25/39, 64%) had been working in obstetrics for between 5 and 10 years. Of the 39 trainees who participated in the evaluation, 30 were female and the majority had received no prior formal training in the caring for families who experienced either antenatal stillbirth (31/39, 80%) or intrapartum stillbirth (34/39, 87%). This was despite almost three quarters of trainees (29/39, 74%) having had direct involvement with one or more stillbirths. Of those who had received training, they cited medical school education and attendance at an Irish Hospice Foundation (national charity that addresses dying, death and bereavement in Ireland; https://hospicefoundation.ie/) Course

**Table 4. Participant demographics.**

| | N = 39 (%) |
|---|---|
| Sex | |
| • Male | 8 (20.5) |
| • Female | 30 (76.9) |
| • Not answered | 1 (2.6) |
| Years in Obstetrics | |
| • <5 | 9 (23.1) |
| • 5–10 | 25 (64.1) |
| • >10 | 4 (10.3) |
| • Not answered | 1 (2.6) |
| Training in Stillbirth Management | |
| • Yes | 7 (17.9) |
| • No | 31 (79.5) |
| • Not answered | 1 (2.6) |
| Training in Intrapartum Death Management | |
| • Yes | 4 (10.3) |
| • No | 34 (87.2) |
| • Not answered | 1 (2.6) |

called "Delivering Bad News" in their respective maternity units as their only training in this area [26]. A full breakdown of participant demographics is presented in Table 4.

A paired-sample t-test was conducted to evaluate whether or not attendance at the workshop improved trainees' confidence in the following key skills and attributes with respective to stillbirth management; 1) breaking bad news, 2) communicating clearly with the family when breaking bad news, 3) communicating empathetically with the family when breaking bad news, 4) recognising the emotional needs of the family, 5) recognising their own emotional responses and 6) supporting their colleagues. Consistently, there was a statistically significant improvement in trainee's level of confidence in all of the above skills after they had attended the workshop. The individual t-statistic for each skill or attribute is presented in Table 5. Communication skills and awareness of their own emotional responses where the areas that appeared to improve the most.

Trainees were overall very positive about the course content and most reported that each of the attributes needed when caring for families who experienced a stillbirth were covered either 'just right' or 'a lot' within the workshop. A detailed breakdown of these findings is presented in Table 6. When asked whether they would recommend the workshop to a colleague, 35/39

**Table 5. Pre and post workshop improvement in confidence in key skills in stillbirth management.**

| Skill | Pre and Post Workshop Mean Scores | Paired Samples t- Test | | | |
|---|---|---|---|---|---|
| | | Mean | 95% CI | t value (df) | significance |
| Breaking bad news | 2.67, 2.95 | 0.282 | 0.086–0.478 | 2.913 (38) | 0.006 |
| Communicating clearly | 2.59, 3.05 | 0.462 | 0.240–0.683 | 4.224 (38) | <0.001 |
| Communicating empathetically | 2.85, 3.18 | 0.333 | 0.162–0.505 | 3.929 (38) | <0.001 |
| Emotional needs of family | 2.72, 3.00 | 0.282 | 0.072–0.492 | 2.723 (38) | .010 |
| Emotional responses of self | 2.62, 3.05 | 0.436 | 0.215–0.656 | 4.001 (38) | <0.001 |
| Supporting colleagues | 2.74, 3.08 | 0.333 | 0.146–0.520 | 3.606 (38) | 0.001 |

**Table 6. Workshop coverage of key attributes for managing stillbirth.**

| Attribute | Trainee Responses N = 39 (%) | | | | | |
|---|---|---|---|---|---|---|
| | Too Little | Little | Just Right | A lot | Too Much | No answer |
| Self-awareness | 0 | 4 (10.3) | 13 (33.3) | 21 (53.8) | 0 | 1 (2.6) |
| Authenticity | 1 (2.6) | 4 (10.3) | 16 (41) | 16 (41) | 1 (2.6) | 1 (2.6) |
| Own emotional responses | 0 | 2 (5.1) | 14 (35.9) | 22 (56.4) | 0 | 1 (2.6) |
| Role of verbal communication | 0 | 3 (7.7) | 13 (33.3) | 21 (53.8) | 1 (2.6) | 1 (2.6) |
| Role of non-verbal communication | 0 | 0 | 8 (20.5) | 29 (74.4) | 1 (2.6) | 1 (2.6) |
| Active listening | 0 | 4 (10.3) | 14 (35.9) | 18 (46.2) | 2 (5.1) | 1 (2.6) |
| Building of rapport | 1 (2.6) | 7 (17.9) | 14 (35.9) | 15 (38.5) | 1 (2.6) | 1 (2.6) |
| Building resilience | 2 (5.1) | 6 (15.4) | 14 (35.9) | 13 (33.3) | 1 (2.6) | 1 (2.6) |

(90%) of trainees said they would. Just one said they would not recommend the workshop and the remaining three did not answer the question.

In general, responses to the free text comments were all positive. Participants valued the different educational approach taken (i.e. applied drama), with some suggesting that the workshops should take place annually as a forum for communication and self-care. The development of empathy and understanding for bereaved parents, which would ultimately make them better clinicians, was also highlighted. Only one person suggested that more time should be spent breaking bad news and that they found "a lot of the activities as time wasting as those were not relevant". A sample of comments is provided in Box 3.

## Discussion

This study evaluated the impact that participation in a novel educational workshop had on improving trainee obstetricians' awareness of compassion, communication and self-care around the time of stillbirth. Following attendance at a day-long workshop, trainees completed a post-workshop feedback questionnaire. The workshop was very positively received; 90% of participants were willing to recommend it to a colleague. Trainees were happy with the course content and felt it met their a priori expectations of the workshop. They valued the use of applied drama to explore key issues and develop important skills which would benefit both them and their patients. We identified a subjective improvement in some of the key skills that obstetricians must have when caring and communicating with parents who are bereaved. We also identified a subjective improvement in how trainees recognise their own emotional responses and support their colleagues in the aftermath of a perinatal death. Our findings add

---

**Box 3. Some of the comments from the workshop participants.**

"A welcome change from didactic lectures, a fresh approach"

"The workshop helps to put ourselves in the patient's position sometimes which makes us better clinicians"

"We should have this sort of training every year to help communicate our own difficulties and relieve stress"

---

to the evidence regarding the effectiveness of applied drama in medical education [17–21] and how patients can be meaningfully involved in postgraduate education efforts [22].

Repeatedly, in the published Irish literature obstetricians have commented on the lack of formal education and training they receive in caring for bereaved parents as well as in caring for themselves in the aftermath of a stillbirth [2, 3]. In addition to this, multiple reviews into specific perinatal mortalities in Ireland, have identified poor communication skills as one of the key shortcomings in the care of bereaved parents [27–29]. The findings reported in our study are therefore of high importance. Adequate, appropriate and stimulating education and training in stillbirth care and self-care is clearly needed to improve patient care in Ireland, and this workshop may be the way to achieve this change. Our findings also have relevance beyond Ireland, with international studies also highlighting the lack of training for obstetricians around supporting parents in the aftermath of a stillbirth, or self-care skills [6, 11].

The main strength of this evaluation is the mandatory requirement for trainees on the RCPI higher specialist training scheme in Obstetrics and Gynaecology to attend this workshop, thereby reducing the likelihood of attender bias and increases the likelihood that trainees with varying levels of experience and interest in stillbirth would attend and give honest feedback. There are some limitations to this study, including the self-reported pre-and post-work assessment of communication and support skills, completed following participation in the workshops. Furthermore, relying on subjective reporting of changes in confidence in the key skills required for effective stillbirth management does not offer definitive proof that this suggested improvement will be translated into clinical practice. A study that objectively shows an improvement in communication, compassion and self-care would be difficult to design and implement. One potential way to show a translation into clinical practice may be to interview trainees several months after the workshop and ask if they have put their skills into practice.

This study has clearly shown that it is an acceptable and appropriate way of training in the areas of stillbirth and self-care in Ireland. Obstetricians in in other countries, not just in Ireland, are affected by stillbirth [6, 30, 31]. The impact of stillbirth does not discriminate based on where an obstetrician lives and works, and as such our findings may be transferable to obstetricians working outside of Ireland. Internationally, there is a marked deficit of studies that evaluate specific educational and self-care interventions for obstetricians to access. Published research has examined the use of both Balint training and reflective writing workshops to address the problem of burnout in obstetrics; none of these intervention types have yet proven to be of benefit [32–34]. Our study definitively shows an improvement in self-reported outcome measures.

In conclusion, this novel way of educating trainee obstetricians, using applied drama techniques, was associated with a self-reported improvement in communication and self-care skills and was very positively received. We therefore recommend that this type of training should be incorporated into postgraduate curriculums in Obstetrics and Gynaecology. In Ireland, since this evaluation was conducted, the workshop has been incorporated into the national postgraduate curriculum in Obstetrics in Ireland [24]. A play—'Singing in the Night' by Tara McKevitt—exploring how parents and doctors cope with late stage stillbirth was also written and performed based on the narratives of bereaved parents and obstetricians that informed the educational workshop, the name of which was inspired by a trainee's performance during one of the workshops.

## Supporting information

**S1 Appendix. Workshop evaluation questionnaire.**
(DOCX)

## Acknowledgments

We would like to sincerely thank all the patients and obstetricians who kindly donated their time and stories which were instrumental in the development of these educational workshops. We also wish to thank Bridgid Doherty (Patient Focus), Cathriona Molloy (independent patient advocate), Tara McKevitt (Playwright), and all study participants.

## Author Contributions

**Conceptualization:** Karen McNamara, Aisling Smith, Mairie Cregan, Lucia Prihodova, Ann O'Shaughnessy, Aisling Martin, Chris Fitzpatrick, Keelin O'Donoghue.

**Data curation:** Karen McNamara, Aisling Smith.

**Formal analysis:** Karen McNamara, Aisling Smith, Lucia Prihodova, Keelin O'Donoghue.

**Investigation:** Karen McNamara.

**Methodology:** Karen McNamara, Aisling Smith, Mairie Cregan, Lucia Prihodova, Ann O'Shaughnessy, Aisling Martin, Chris Fitzpatrick, Keelin O'Donoghue.

**Project administration:** Karen McNamara, Keelin O'Donoghue.

**Supervision:** Keelin O'Donoghue.

**Writing – original draft:** Karen McNamara, Keelin O'Donoghue.

**Writing – review & editing:** Karen McNamara, Aisling Smith, Brid Shine, Mairie Cregan, Lucia Prihodova, Ann O'Shaughnessy, Aisling Martin, Jenny Macdonald, Phil Kingston, Chris Fitzpatrick, Marita Hennessy, Keelin O'Donoghue.

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
