## [Decision Letter · Decision Letter 0]

6 Jul 2022

PONE-D-22-07345

Addressing obstetricians’ awareness of compassion, communication and self-care when caring for families experiencing stillbirth

PLOS ONE

Dear Dr. Hennessy,

Thank you for submitting your manuscript to PLOS ONE. After careful consideration, we feel that it has merit but does not fully meet PLOS ONE’s publication criteria as it currently stands. Therefore, we invite you to submit a revised version of the manuscript that addresses the points raised during the review process.

The reviewers have positive feedback about this manuscript. Yet, two of them suggested important revisions that I fully support to further improve the quality of the study-presentation. Many thanks for addressing the proposed revisions in detail. 

We look forward to receiving your revised manuscript.

Kind regards,

Sara Rubinelli

Academic Editor

PLOS ONE

https://journals.plos.org/plosone/s/file?id=ba62/PLOSOne_formatting_sample_title_authors_affiliations.pdf".

“This study was completed in part in the Irish Centre for Fetal and Neonatal Translational Research (now known as The Irish Centre for Maternal and Child Health) and as such was in part funded by Science Foundation Ireland (12/RC/2722).”

“The author(s) received no specific funding for this work”

3. We noted in your submission details that a portion of your manuscript may have been presented or published elsewhere. [This work formed part of KMN’s doctoral studies. A previous version of this manuscript is presented as a pre-print within her doctoral thesis, available on CORA, the Cork Open Research Archive, at University College Cork; KMN retains copyright to reuse the material in the present manuscript. Available at: https://cora.ucc.ie/handle/10468/9626] Please clarify whether this [conference proceeding or publication] was peer-reviewed and formally published. If this work was previously peer-reviewed and published, in the cover letter please provide the reason that this work does not constitute dual publication and should be included in the current manuscript.

4. In your Data Availability statement, you have not specified where the minimal data set underlying the results described in your manuscript can be found. PLOS defines a study's minimal data set as the underlying data used to reach the conclusions drawn in the manuscript and any additional data required

to replicate the reported study findings in their entirety. All PLOS journals require that the minimal data set be made fully available. For more information about our data policy, please see http://journals.plos.org/plosone/s/data-availability.

Reviewers' comments:

Reviewer's Responses to Questions

**Comments to the Author**

1. Is the manuscript technically sound, and do the data support the conclusions?

Reviewer #1: Partly

Reviewer #2: Partly

Reviewer #3: Yes

2. Has the statistical analysis been performed appropriately and rigorously? 

Reviewer #1: I Don't Know

Reviewer #2: Yes

Reviewer #3: Yes

3. Have the authors made all data underlying the findings in their manuscript fully available?

Reviewer #1: Yes

Reviewer #2: No

Reviewer #3: Yes

4. Is the manuscript presented in an intelligible fashion and written in standard English?

Reviewer #1: Yes

Reviewer #2: Yes

Reviewer #3: Yes

5. Review Comments to the Author

Reviewer #1: I really enjoyed reading this piece of work. It describes a really innovative approach to learning in a vital part of maternity care.

The paper is written well and is easy to read.

With regards to the methodology I believe the delegates completed the questionnaire at the end of the session, including a self-analysis of their knowledge before the training. I wonder if asking this after the training influenced these scores at all?

I can’t comment on the robustness of the methodology or the statistical interpretation as outside of my area of expertise.

As a clinician working in the field of maternity safety I welcome this type of innovative training for situations that are less commonly encountered. I also think the exploration of the affect of stillbirth on obstetricians is an important aspect of this work.

Reviewer #2: McNamara et al present a compelling, important manuscript about an understudied and underappreciated area of obstetrical care: how to best care for patients with late trimester pregnancy or early neonatal losses. The paper is straightforward, clearly written, with results that are concisely laid out.

That being said, there is one large omission: what actually was the training that the authors are discussing? One key element of any publication, and this especially holds for the PLOS journals, is to share all the data that would be necessary for a reader to repeat and confirm the described studies in another location, at another time, with a new group of test subjects. Unfortunately, there are no details presented on what actually happened in the workshops that were held or how someone might repeat such a workshop.

As with any scientific study the methods need to be a clear set of instructions on how to repeat the study. It is not obvious to me how this might be done, but some suggestions follow that might be presented in the Appendix:

1. Can the authors supply transcripts or audio recordings of the workshops? Might these include the instructions given to the participants, the exercises they participated in, the actual mock loss sessions they had to act in? In the spirit of Sir Alec Guinness’ quote, can they supply the “movie script” of the workshop?

2. Do the authors have any video of any of the workshops? If so, can these be posted, along with the script so others can actually emulate and replicate what this manuscript is describing?

3. If there are no transcripts, audio or video recordings (which would be unfortunate), can the authors recreate a mock workshop in writing that has sufficient detail to act as a lesson plan for someone trying to enact this program in their clinic or hospital?

Reviewer #3: Title: Consider making a shorter title. Use of the Oxford Comma in the Title

Abstract: capitalize National Theatre of Ireland and other Institutions. Use of the oxford comma (lines 59, 64).

Introduction: References should be in order of apparition (where is reference 15?). Standarize use of brackets vs parenthesis.

Line 127: Which other national advocacy service, what has hapenned to Patient Focus since 2018

Methods: Abstain from describing opinions or results in the methodology.

Results: Line 280: how many trainees left early or had other commitments

More information to reproduce your workshop in other places is needed.

6. PLOS authors have the option to publish the peer review history of their article (what does this mean?). If published, this will include your full peer review and any attached files.

Reviewer #1: No

Reviewer #2: No

Reviewer #3: No

---

## [Author Response · Author response to Decision Letter 0]

28 Jul 2022

GENERAL COMMENTS

COMMENT: Thank you for submitting your manuscript to PLOS ONE. After careful consideration, we feel that it has merit but does not fully meet PLOS ONE’s publication criteria as it currently stands. Therefore, we invite you to submit a revised version of the manuscript that addresses the points raised during the review process. The reviewers have positive feedback about this manuscript. Yet, two of them suggested important revisions that I fully support to further improve the quality of the study-presentation. Many thanks for addressing the proposed revisions in detail. 

RESPONSE: We thank all three reviewers and the Editor for taking the time to review our manuscript and for their considered and constructive feedback. We believe that the manuscript is much improved as a result. We have addressed each reviewer’s comments, separately, below. Revised/added text is highlighted in red (in response to reviewers uploaded in previous section). We have tracked changes within the revised manuscript.

COMMENT: When submitting your revision, we need you to address these additional requirements.

https://journals.plos.org/plosone/s/file?id=ba62/PLOSOne_formatting_sample_title_authors_affiliations.pdf".

RESPONSE:

We have checked our manuscript against the PLOS ONE style requirements and made revisions where necessary.

COMMENT:

“This study was completed in part in the Irish Centre for Fetal and Neonatal Translational Research (now known as The Irish Centre for Maternal and Child Health) and as such was in part funded by Science Foundation Ireland (12/RC/2722).”

We note that you have provided funding information that is not currently declared in your Funding Statement. However, funding information should not appear in the Acknowledgments section or other areas of your manuscript. We will only publish funding information present in the Funding Statement section of the online submission form. Please remove any funding-related text from the manuscript and let us know how you would like to update your Funding Statement. Currently, your Funding Statement reads as follows: “The author(s) received no specific funding for this work”

RESPONSE: Sincere apologies for this error. We have deleted the funding section from the manuscript and include an amended funding statement in our associated cover letter, as follows: “This study was completed in part in the Irish Centre for Fetal and Neonatal Translational Research (now known as The Irish Centre for Maternal and Child Health) and as such was in part funded by Science Foundation Ireland (12/RC/2722).”

COMMENT: 3. We noted in your submission details that a portion of your manuscript may have been presented or published elsewhere. [This work formed part of KMN’s doctoral studies. A previous version of this manuscript is presented as a pre-print within her doctoral thesis, available on CORA, the Cork Open Research Archive, at University College Cork; KMN retains copyright to reuse the material in the present manuscript. Available at: https://cora.ucc.ie/handle/10468/9626] Please clarify whether this [conference proceeding or publication] was peer-reviewed and formally published. If this work was previously peer-reviewed and published, in the cover letter please provide the reason that this work does not constitute dual publication and should be included in the current manuscript.

RESPONSE: We can confirm that the preprint of this manuscript was published within KMN’s doctoral thesis. While available within the limits of the Cork Open Research Archive at University College Cork, it was not a paper and never published as such, i.e. a manuscript officially peer reviewed and published. Furthermore, the version of the manuscript submitted to PLOS One significantly differs to the preprint. For these reasons, this work does not constitute dual publication and should be included in the current manuscript.

COMMENT: 4. In your Data Availability statement, you have not specified where the minimal data set underlying the results described in your manuscript can be found. PLOS defines a study's minimal data set as the underlying data used to reach the conclusions drawn in the manuscript and any additional data required to replicate the reported study findings in their entirety. All PLOS journals require that the minimal data set be made fully available. For more information about our data policy, please see http://journals.plos.org/plosone/s/data-availability.

RESPONSE: We have deleted the data availability statement from the manuscript and include an amended statement in our associated cover letter, as follows: “Data cannot be shared publicly because it contains sensitive information about the participants and they did not provide consent for public data sharing. The current approval by the Research Ethics Committee at the Royal College of Physicians of Ireland (RCPI RECSAF 71) does not include data sharing. A minimal data set could be shared by request for the sole purpose of replicating the present study. Data requests may be sent to the senior author (KOD; k.odonoghue@ucc.ie) or the Research Ethics Committee at the Royal College of Physicians of Ireland (research@rcpi.ie).”

COMMENT: 5. We note that you have indicated that data from this study are available upon request. PLOS only allows data to be available upon request if there are legal or ethical restrictions on sharing data publicly. For more information on unacceptable data access restrictions, please see http://journals.plos.org/plosone/s/data-availability#loc-unacceptable-data-access-restrictions.

RESPONSE: As detailed above

REVIEWER #1

COMMENT: I really enjoyed reading this piece of work. It describes a really innovative approach to learning in a vital part of maternity care. The paper is written well and is easy to read. I can’t comment on the robustness of the methodology or the statistical interpretation as outside of my area of expertise. As a clinician working in the field of maternity safety I welcome this type of innovative training for situations that are less commonly encountered. I also think the exploration of the affect of stillbirth on obstetricians is an important aspect of this work.

RESPONSE: Thank you for taking the time to review our manuscript and for your positive comments on this work.

COMMENT: With regards to the methodology I believe the delegates completed the questionnaire at the end of the session, including a self-analysis of their knowledge before the training. I wonder if asking this after the training influenced these scores at all?

RESPONSE: Thank you for this comment; this is certainly possible. We have noted the subjective nature of the assessment within the manuscript: “We identified a subjective improvement in some of the key skills that obstetricians must have when caring and communicating with parents who are bereaved. We also identified a subjective improvement in how trainees recognise their own emotional responses and support their colleagues in the aftermath of a perinatal death.” [Lines 376-381]. 

Further to your comment, we have added more details to the limitations section, which now reads: “There are some limitations to this study, including the self-reported pre-and post-work assessment of communication and support skills, completed following participation in the workshops. Furthermore, relying on subjective reporting of changes in confidence in the key skills required for effective stillbirth management does not offer definitive proof that this suggested improvement will be translated into clinical practice.” [Lines 402-407]

REVIEWER #2:

COMMENT: McNamara et al present a compelling, important manuscript about an understudied and underappreciated area of obstetrical care: how to best care for patients with late trimester pregnancy or early neonatal losses. The paper is straightforward, clearly written, with results that are concisely laid out.

RESPONSE: Thank you for your time and expertise in reviewing our manuscript.

COMMENT: That being said, there is one large omission: what actually was the training that the authors are discussing? One key element of any publication, and this especially holds for the PLOS journals, is to share all the data that would be necessary for a reader to repeat and confirm the described studies in another location, at another time, with a new group of test subjects. Unfortunately, there are no details presented on what actually happened in the workshops that were held or how someone might repeat such a workshop.

RESPONSE: Thank you for raising this point. We discussed how much detail we should, and could, provide when preparing this manuscript. In the ‘materials and methods’ section – specifically ‘workshop development and delivery’, we endeavour to describe what the workshops entailed. We outline the aims and objectives, and the writing, acting, movement and reflective activities undertaken (Box 2). Importantly we note that “While the aims and objectives were the same for each workshop, the exact content of each workshop differed somewhat each time, as trainees differed in terms of their experiences and inputs/level of interaction, and facilitators also evolved after engaging in and reflecting on each workshop.” [Lines 180-184]. We have added a workshop outline to the manuscript (Table 2, lines 225-236; Table numbers have been updated accordingly) which provides more insight into what happened in the workshops.

COMMENT: As with any scientific study the methods need to be a clear set of instructions on how to repeat the study. It is not obvious to me how this might be done, but some suggestions follow that might be presented in the Appendix.

RESPONSE: Thank you for these suggestions; we have addressed each separately below.

COMMENT: 1. Can the authors supply transcripts or audio recordings of the workshops? Might these include the instructions given to the participants, the exercises they participated in, the actual mock loss sessions they had to act in? In the spirit of Sir Alec Guinness’ quote, can they supply the “movie script” of the workshop?

RESPONSE: We did not video/audio-record any of the workshops. Due to the nature of the workshops they were not recorded, and indeed any external in-person observation was avoided, to enable trainees to feel comfortable in participating, and sharing their experiences, fully.

COMMENT: 2. Do the authors have any video of any of the workshops? If so, can these be posted, along with the script so others can actually emulate and replicate what this manuscript is describing?

RESPONSE: As noted above; we did not record the workshops.

COMMENT: 3. If there are no transcripts, audio or video recordings (which would be unfortunate), can the authors recreate a mock workshop in writing that has sufficient detail to act as a lesson plan for someone trying to enact this program in their clinic or hospital?

RESPONSE: As noted above, we provide details of the aims and objectives, and the writing, acting, movement and reflective activities undertaken (Box 2). The exact content of each workshop differed somewhat each time, as trainees differed in terms of their experiences and inputs/level of interaction, and facilitators also evolved after engaging in and reflecting on each workshop. We have added a workshop outline (Table 2).

REVIEWER #3

COMMENT: Title: Consider making a shorter title

RESPONSE: The title was previously:

Addressing obstetricians’ awareness of compassion, communication and self-care when caring for families experiencing stillbirth: Evaluation of a novel educational workshop using applied drama techniques, developed in collaboration with a national theatre, training body and patient groups

The title now reads:

“Addressing obstetricians’ awareness of compassion, communication, and self-care when caring for families experiencing stillbirth: Evaluation of a novel educational workshop using applied drama techniques”

COMMENT: Use of the Oxford Comma in the Title

RESPONSE: This has been added, see above

COMMENT: Abstract: capitalize National Theatre of Ireland and other Institutions. Use of the oxford comma (lines 59, 64).

RESPONSE: We have capitalised National Theatre of Ireland; others have not been capitalised as they are not the official names of institutions. This sentence now reads:

“We developed a novel workshop for trainee obstetricians using applied drama techniques–in collaboration with the National Theatre of Ireland, the national training body for obstetricians and gynaecologists, and patient support groups–to teach obstetricians skills in communication and self-care around the time of stillbirth.” [Lines 31-35]

We have added the Oxford comma to lines 54 and 59.

They now read:

“Following the workshop there was a significant improvement in trainee’s level of confidence in breaking bad news, communicating clearly with the family when breaking bad news, recognising the emotional needs of the family, recognising their own emotional responses, and supporting their colleagues.” [Lines 50-54]

“Adequate, appropriate, and stimulating education and training in stillbirth care and self-care is clearly needed to improve patient care.” [Lines 59-60]

COMMENT: Introduction: References should be in order of apparition (where is reference 15?). Standarize use of brackets vs parenthesis.

RESPONSE: Reference 15 appears on Line 97, as follows:

“In Ireland, the importance of empathy and compassion in bereavement care has been highlighted in the National Maternity Strategy, National Standards for Safer Better Maternity Services and National Standards for Bereavement Care in the maternity services [14-16].” We have checked the order of references and have standardised in-text citations according to the Journal’s requirements [Vancouver; brackets]

COMMENT: Line 127: Which other national advocacy service, what has hapenned to Patient Focus since 2018

RESPONSE: On July 1st 2018, the patient advocacy services, previously provided by Patient Focus, moved to Sage Advocacy which is now a support and advocacy service for vulnerable adults, older people and healthcare patients. Further information available at: https://www.sageadvocacy.ie/about

This line now reads:

“Until July 2018, Patient Focus was one of the country’s leading patient advocacy services - before its merging with another national advocacy service, Sage Advocacy.” [Lines 120-122]

COMMENT: Methods: Abstain from describing opinions or results in the methodology.

RESPONSE: Thank you for this comment. In the methodology we provide information on both the design and development of the workshops as well as the evaluation methodology. The former may contain opinions or results/outcomes to provide the reader with greater insight into the content/delivery of the workshops – these do not relate to the evaluation of the workshops which is the focus of this paper.

COMMENT: Results: Line 280: how many trainees left early or had other commitments

RESPONSE: Twenty trainees left early or had other commitments, noted as follows: 

“Of those who attended, 39/59 (66%) completed the workshop evaluation questionnaires. The main reason for non-participation in the evaluation was trainees leaving immediately after the workshop, or slightly earlier, due to other commitments”. [Lines 295-298]

---

## [Decision Letter · Decision Letter 1]

28 Oct 2022

Addressing obstetricians’ awareness of compassion, communication, and self-care when caring for families experiencing stillbirth: Evaluation of a novel educational workshop using applied drama techniques

PONE-D-22-07345R1

Dear Dr. Hennessy,

We’re pleased to inform you that your manuscript has been judged scientifically suitable for publication and will be formally accepted for publication once it meets all outstanding technical requirements.

Kind regards,

Sara Rubinelli

Academic Editor

PLOS ONE

Additional Editor Comments (optional):

Reviewers' comments:

Reviewer's Responses to Questions

**Comments to the Author**

1. If the authors have adequately addressed your comments raised in a previous round of review and you feel that this manuscript is now acceptable for publication, you may indicate that here to bypass the “Comments to the Author” section, enter your conflict of interest statement in the “Confidential to Editor” section, and submit your "Accept" recommendation.

Reviewer #1: All comments have been addressed

Reviewer #2: (No Response)

Reviewer #3: All comments have been addressed

2. Is the manuscript technically sound, and do the data support the conclusions?

Reviewer #1: Yes

Reviewer #2: No

Reviewer #3: Yes

3. Has the statistical analysis been performed appropriately and rigorously? 

Reviewer #1: I Don't Know

Reviewer #2: I Don't Know

Reviewer #3: Yes

4. Have the authors made all data underlying the findings in their manuscript fully available?

Reviewer #1: Yes

Reviewer #2: No

Reviewer #3: Yes

5. Is the manuscript presented in an intelligible fashion and written in standard English?

Reviewer #1: Yes

Reviewer #2: Yes

Reviewer #3: Yes

6. Review Comments to the Author

Reviewer #1: Thank you for addressing my previous comments. No new comments.

Reviewer #2: (No Response)

Reviewer #3: This is an interesting study, the paper is generally well written and structured.

The authors have addressed all of my comments and made the appropiate corrections.

7. PLOS authors have the option to publish the peer review history of their article (what does this mean?). If published, this will include your full peer review and any attached files.

Reviewer #1: No

Reviewer #2: No

Reviewer #3: **Yes: **Francisco Javier Ruiloba Portilla

---

## [Editor Report · Acceptance letter]

8 Nov 2022

PONE-D-22-07345R1 

Addressing obstetricians’ awareness of compassion, communication, and self-care when caring for families experiencing stillbirth: Evaluation of a novel educational workshop using applied drama techniques 

Dear Dr. Hennessy:

I'm pleased to inform you that your manuscript has been deemed suitable for publication in PLOS ONE. Congratulations! Your manuscript is now with our production department. 

Kind regards, 

on behalf of

Dr. Sara Rubinelli 

Academic Editor

PLOS ONE